# Decoupling the "What" and "Where" with Polar Coordinate Positional Embedding

## Abstract

The attention mechanism in a Transformer architecture matches key to query based on both content—the *what*—and position in a sequence—the *where*. We present an analysis indicating that what and where are entangled in the popular rotary position embedding (RoPE). This entanglement can impair performance particularly when decisions require independent matches on these two factors. We propose an improvement to RoPE, which we call *Polar Coordinate Position Embedding* or *PoPE*, that eliminates the what-where confound. PoPE is far superior on a diagnostic task requiring indexing solely by position or by content. On autoregressive sequence modeling in music, genomic, and natural language domains, Transformers using PoPE as the positional encoding scheme outperform baselines using RoPE with respect to evaluation loss (perplexity) and downstream task performance. On language modeling, these gains persist across model scale, from 124M to 774M parameters. Crucially, PoPE shows strong zero-shot length extrapolation capabilities, whereas RoPE's performance degrades significantly on longer sequences at test time without fine tuning or the use of position-interpolation methods.

## 1 Introduction

In the prehistory of deep learning, a key challenge was representing sequential or position-coded data. Tasks of interest included recognizing words from letter orderings (McClelland & Rumelhart, 1981), recognizing speech from power spectra (Waibel et al., 1989), compressing long sequences with time-lags between predictable events (Schmidhuber et al., 1993), classifying hand-drawn symbols from strokes (Yaeger, 1996), and forecasting time series from samples (Mozer, 1993).

In the 1980s, two approaches were common: recurrent neural nets (RNNs) and slot-based encodings. RNNs are a means of encoding sequence position implicitly in the ordering of inputs. By contrast, slot-based encodings partition an input vector into separate components for each sequence step. A significant advantage of RNNs is their approximate *translation equivariance*, meaning that shifting the absolute position of a sub-sequence in the input yields analogous shifts in the model state. Slot-based representations do not share this property: learning to respond to content in one slot does not generalize to the same content in other slots.

The Transformer architecture (Vaswani et al., 2017) changed the game for slot-based representations. Through its self-attention mechanism, the basic Transformer is not only translation equivariant but also translation and permutation *invariant* (subject to causal attention boundaries). The challenge with Transformers thus became how to obtain translation equivariance without losing sensitivity to the relative positions of input elements. The solutions that emerged involve enriching latent representations to encode not only their content—the *what*—but also the relationship between their sequence positions—the *where* (Vaswani et al., 2017; Dai et al., 2019; Su et al., 2024). These solutions rightly assume that both content and position are essential to modeling complex sequences like language.

In this article, we argue that the Rotary Position Embedding (RoPE) (Su et al., 2024), a widely adopted solution, entangles the what and where in a way that can impair model performance, particularly when decision making requires independent matches on these two factors. We propose an alternative technique, which we call PoPE, that shares the advantages of RoPE over other positional encodings while allowing the Transformer to implement rules in which the key-query

match can be characterized as a conjunction of a what match and a where match. Although PoPE is only a minor modification of RoPE, it introduces a powerful inductive bias that improves data efficiency, asymptotic accuracy, and yields superior context-length generalization.

## 2 BACKGROUND

RoPE (Su et al., 2024) is the dominant approach to incorporate positional information in many frontier language models, including Llama 3 (Grattafiori et al., 2024), DeepSeekv3 (Liu et al., 2024), Gemma 3 (Team et al., 2025), and Qwen3 (Yang et al., 2025). It produces an attention score for each query-key pair that is based on both how well they match and their relative positions in the input sequence.

To explain RoPE, consider a specific attention head in a specific layer which performs a match between a query in position $t$, denoted $\boldsymbol{q}_t$, and a key in position $s$, denoted $\boldsymbol{k}_s$. (If it helps to remember notation, $t$ and $s$ could denote t̲arget and s̲ource of attention, respectively.) The key and query are $d$-dimensional vectors that are partitioned into $d/2$ two-dimensional components. We denote component $c \in \{1, \ldots, d/2\}$ of query and key by $\boldsymbol{q}_{tc}$ and $\boldsymbol{k}_{sc}$, respectively. RoPE first rotates each component $c$ in the 2D plane by an angle proportional to the key and query positions. If $\boldsymbol{R}(\phi)$ is a $2 \times 2$ matrix that performs a rotation by angle $\phi$, the rotated query and key are $\boldsymbol{R}(t\theta_c)\,\boldsymbol{q}_{tc}$ and $\boldsymbol{R}(s\theta_c)\,\boldsymbol{k}_{sc}$, where $\theta_c$ is a component-specific , i.e. $\theta_c = \theta^{-2(c-1)/d} : c = 1, \ldots, d/2$ and $\theta$ is called the base wavelength. The formation of query (or key) components and their rotation in 2D are depicted on the left side of Figure 1.

The corresponding components of key and query are matched via a dot product and summed to obtain an attention score:

$$a_{ts}^{\text{RoPE}} = \sum_{c=1}^{d/2} \left[\boldsymbol{R}(t\theta_c)\,\boldsymbol{q}_{tc}\right]^{\text{T}} \left[\boldsymbol{R}(s\theta_c)\,\boldsymbol{k}_{sc}\right] = \sum_{c=1}^{d/2} \boldsymbol{q}_{tc}^{\text{T}}\,\boldsymbol{R}((s-t)\theta_c)\,\boldsymbol{k}_{sc} . \tag{1}$$

The rotation to bring components into alignment depends only on the relative positions of the key and query, not their absolute positions.

Our perspective on RoPE begins by re-expressing the key and query components from Cartesian to polar coordinates, $\boldsymbol{k}_{sc} \Leftrightarrow (\mu_{k_{sc}}, \phi_{k_{sc}})$ and $\boldsymbol{q}_{tc} \Leftrightarrow (\mu_{q_{tc}}, \phi_{q_{tc}})$, where

$$\boldsymbol{k}_{sc} = \boldsymbol{R}(\phi_{k_{sc}}) \begin{bmatrix} \mu_{k_{sc}} \\ 0 \end{bmatrix} \quad \text{and} \quad \boldsymbol{q}_{tc} = \boldsymbol{R}(\phi_{q_{tc}}) \begin{bmatrix} \mu_{q_{tc}} \\ 0 \end{bmatrix}.$$

With this notation, we can compose the rotations and write the attention score as

$$a_{ts}^{\text{RoPE}} = \sum_{c=1}^{d/2} [\mu_{q_{tc}} \quad 0]\,\boldsymbol{R}((s-t)\theta_c - \phi_{q_{tc}} + \phi_{k_{sc}}) \begin{bmatrix} \mu_{k_{sc}} \\ 0 \end{bmatrix}$$

$$= \sum_{c=1}^{d/2} \mu_{q_{tc}} \mu_{k_{sc}} \cos((s-t)\theta_c + \phi_{k_{sc}} - \phi_{q_{tc}}). \tag{2}$$

This algebra makes clear that each two-element component of the embedding is transformed into a single magnitude and also introduces, via $\phi_{q_{tc}}$ and $\phi_{k_{sc}}$, an adjustment to the relative position (phase) that yields the maximal response. Thus, both the key and the query confound information about the presence or absence of features (the 'what') and relative positions (the 'where'). Our hypothesis is that we can improve model performance by disentangling these two distinct sorts of information, specifically by removing the interaction term $\phi_{k_{sc}} - \phi_{q_{tc}}$.

## 3 METHOD

In RoPE, we interpreted the $d/2$ components of the key and query as complex numbers. In the method we propose, we utilize an alternative form of polar-coordinate representation of key and query. We refer to our method as *PoPE*, for *P̲olar C̲oordinate P̲ositional E̲mbedding*. In PoPE, we transform the key and query into corresponding $d$-element complex vectors, which we refer to as

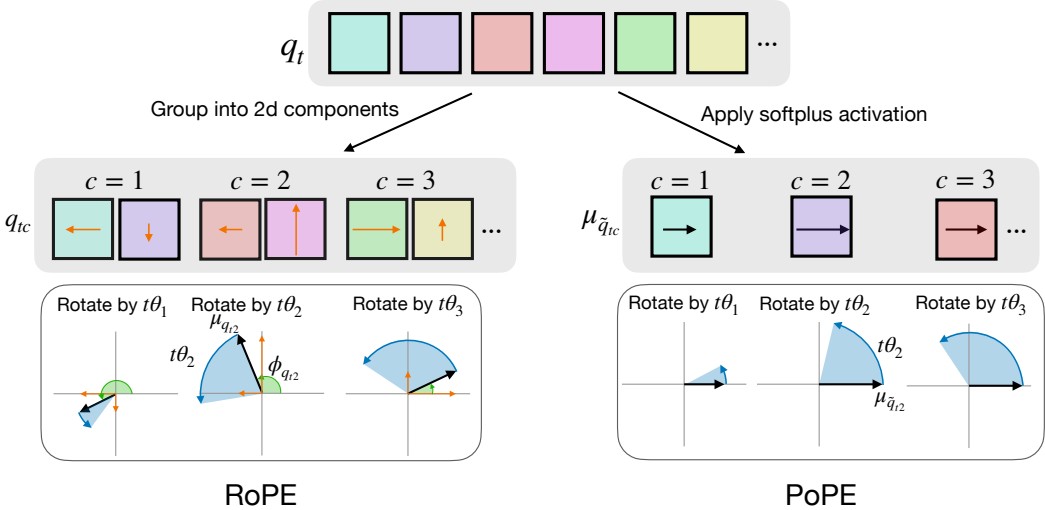

Figure 1: Illustration compares how RoPE and PoPE encode relative positions via rotations of queries. Left: Three complex-valued RoPE components having magnitudes $\mu_{q_{tc}}$ (black arrows) and initial phases $\phi_{q_{tc}}$ (green arcs) are constructed from three pairs of embedding features (orange arrows) of the query vector $q_t$ (gray box) at sequence position $t$. These RoPE components are then rotated by angles $t\theta_c$ (blue arcs). Right: Three magnitude components $\mu_{q_{tc}}$ (black arrows) of complex-valued PoPE components are constructed from three embedding features of the query vector $q_t$ (gray box) by applying softplus activation. These magnitudes (complex numbers with zero phases) are then rotated by angles $t\theta_c$ (blue arcs). PoPE uses twice the number of components than RoPE as it applies rotations to each component of the query vector $q_t$.

$\tilde{k}_s$ and $\tilde{q}_t$. In each, the magnitude of element $c \in \{1, \ldots, d\}$ is a rescaling of the corresponding element of the original real-valued key or query:

$$\mu_{\tilde{k}_{sc}} = \sigma(k_{sc}) \quad \text{and} \quad \mu_{\tilde{q}_{tc}} = \sigma(q_{tc}), \tag{3}$$

where $\sigma(x) = \ln(1 + e^x)$ denotes the softplus activation function. The softplus ensures the $\mu$ are non-negative, permitting them to be interpreted as magnitudes. The phases are position dependent:

$$\phi_{\tilde{k}_{sc}} = s\theta_c \quad \text{and} \quad \phi_{\tilde{q}_{tc}} = t\theta_c, \tag{4}$$

where $\theta_c$ is a component-specific frequency, i.e. $\theta_c = \theta^{(c-1)/d} : c = 1, \ldots, d$. With this definition of the key $\tilde{k}_s \in \mathbb{C}^d$ and query $\tilde{q}_t \in \mathbb{C}^d$, the attention score can elegantly be defined as:

$$\boldsymbol{a}_{ts}^{\text{PoPE}} = \Re\left[\tilde{q}_t^H \, \tilde{k}_s\right] = \Re\left[\sum_{c=1}^{d} \tilde{q}_{tc}^H \, \tilde{k}_{sc}\right] = \sum_{c=1}^{d} \mu_{\tilde{q}_{tc}} \mu_{\tilde{k}_{sc}} \cos((s-t)\theta_c), \tag{5}$$

where $H$ denotes the conjugate transpose which involves applying complex conjugation to each component. The PoPE attention score (Equation 5) is quite similar in form to the RoPE attention score (Equation 2), except that: i) $c$ is an index over individual elements of the key and query and not over pairs of elements, thereby doubling the number of frequencies from $d/2$ to $d$, and ii) the interaction term causing key and query to influence phase has been eliminated.

Extending this formulation, each attention head might benefit by introducing a learnable but fixed bias term to replace the RoPE interaction term:

$$\boldsymbol{a}_{ts}^{\text{PoPE}} = \sum_{c=1}^{d} \mu_{\tilde{q}_{tc}} \mu_{\tilde{k}_{sc}} \cos((s-t)\theta_c + \delta_c), \tag{6}$$

where $\delta_c \in \mathbb{R}$ is a learnable bias that tunes the optimal relative offset for each frequency $c$. Figure 1 visualizes the different ways by which RoPE and PoPE encode relative positions as rotations applied to queries and keys. There are several ways to initialize the bias terms $\delta_c$. We find that two

good options are to initialize it either with $\delta_c = 0$ or $\delta_c \sim \text{Uniform}(-2\pi, 0)$. Further, we bound $\delta_c$ so that it to always lies in the interval $[-2\pi, 0]$, i.e. $\theta_c = \min(\max(\theta_c, -2\pi), 0)$ and found this improves stability. We find that the zero initialization is important for length generalization while the uniform one gives slightly better in-distribution performance.

**Efficient Implementation.** We implemented PoPE using Triton, starting from the example code for Flash Attention 2 (Dao, 2024).[1] We modify the kernel to take complex-valued keys and queries in Cartesian form and compute the real part of their product inside the kernel, without ever materializing the resulting complex matrix of the query-key dot product. We can compute the real and imaginary components of the complex-valued $\tilde{q}_{tc}$ from its polar form as:

$$x_{\tilde{q}_{tc}} = \mu_{\tilde{q}_{tc}} \cos(\phi_{\tilde{q}_{tc}}) \quad \text{and} \quad y_{\tilde{q}_{tc}} = \mu_{\tilde{q}_{tc}} \sin(\phi_{\tilde{q}_{tc}}), \tag{7}$$

where $x_{\tilde{q}_{tc}}$ and $y_{\tilde{q}_{tc}}$ denote the real and imaginary components of a complex-valued query feature $\tilde{q}_{tc} \in \mathbb{C}$. Similarly, we can obtain the real and imaginary components of $\tilde{k}_{sc}$ also additionally accounting for the learnable phase-shifts $\delta_c$ as:

$$x_{\tilde{k}_{sc}} = \mu_{\tilde{k}_{sc}} \cos(\phi_{\tilde{k}_{sc}} + \delta_c) \quad \text{and} \quad y_{\tilde{k}_{sc}} = \mu_{\tilde{k}_{sc}} \sin(\phi_{\tilde{k}_{sc}} + \delta_c), \tag{8}$$

Then, note that the multiplication of complex numbers in Cartesian form is:

$$\tilde{q}_{tc}^H \tilde{k}_{sc} = (x_{\tilde{q}_{tc}} - iy_{\tilde{q}_{tc}})(x_{\tilde{k}_{sc}} + iy_{\tilde{k}_{sc}}) = x_{\tilde{q}_{tc}} x_{\tilde{k}_{sc}} - i^2 y_{\tilde{q}_{tc}} y_{\tilde{k}_{sc}} + i(x_{\tilde{q}_{tc}} y_{\tilde{k}_{sc}} - y_{\tilde{q}_{tc}} x_{\tilde{k}_{sc}}) \tag{9}$$

The efficient computation of the attention score for complex-valued query $\tilde{\boldsymbol{q}}_t$ and key $\tilde{\boldsymbol{k}}_s$ is:

$$\boldsymbol{a}_{ts}^{\text{PoPE}} = \Re\left[\tilde{\boldsymbol{q}}_t^H \tilde{\boldsymbol{k}}_s\right] = \sum_{c=1}^{d} x_{\tilde{q}_{tc}} x_{\tilde{k}_{sc}} + y_{\tilde{q}_{tc}} y_{\tilde{k}_{sc}} \tag{10}$$

Thus, our custom Flash Attention for PoPE requires only a single additional multiplication compared to the standard one. However, it requires twice the memory usage and bandwidth to store and load the complex-valued keys and values from the global memory. It is possible to avoid this memory overhead by loading the magnitudes of the keys and queries directly and performing the rotation inside the kernel. Now, the only overhead is the additional multiplication. We chose to go with the slower, but general variant, which takes as arguments complex-valued keys and queries in Cartesian form and not perform the memory optimization, since this would prevent us from easy prototyping of different PoPE variants.

## 4 RESULTS

In all experiments, we compare our method, PoPE, to the popular RoPE (Su et al., 2024) scheme using two Transformers with identical model and training hyperparameters, the only difference being their positional encoding schemes. For more details on how the datasets are generated, preprocessed, and tokenized, refer to Appendix A.1. In all experiments, we use a decoder-only Transformer architecture (Vaswani et al., 2017; Radford et al., 2018) with causal masking for autoregressive sequence modeling. The only change applied is the use of RMSNorm (Zhang & Sennrich, 2019) instead of LayerNorm (Ba et al., 2016) for normalization, as is common in the latest frontier models. For more details on the model configuration and training hyperparameters used for each experiment refer to Appendix A.2 and Appendix A.3 respectively.

**Indirect Indexing.** We introduce a task that requires identifying a target character within a variable-length source string. The target is defined to be at a specified relative offset (left or right) from a specified source character. For example, in the input `QEOHoUbKfeSrMVNlCzXu`, `z`, `-3`, `N`; the target `N` is three places to the left of source `z` in the string. Predicting the final (target) character of this sequence requires models to learn to independently manipulate the content and positional information of tokens and to apply pointer arithmetic operations. The dataset is constructed by procedurally generating examples of source strings, source symbols, and relative shifts. We compare RoPE (Su et al., 2024) against PoPE by training two Transformer models with cross-entropy loss applied only on the final (target) token and evaluated on the accuracy of final token. Table 1 shows the mean and standard deviation (3 seeds) in final token accuracy on the test set.

---
[1] https://triton-lang.org/main/getting-started/tutorials/06-fused-attention.html

While RoPE struggles to learn this task and reaches an average accuracy of just 11%, our method PoPE solves the task almost perfectly, reaching an average accuracy of nearly 95%. This result highlights the difficulty that RoPE has in teasing apart 'what' and 'where' information, i.e., identifying the position of certain content and determining the content at a certain relative position. In contrast, PoPE efficiently learns a generalizable solution for the task, presumably because 'what' and 'where' information can be disentangled in key-query matching.

Table 1: Accuracy (with standard deviation) on the test split for the Indirect Indexing task.

| Positional Enc. | Indirect Idx. |
|---|---|
| RoPE | $11.16 \pm 2.45$ |
| PoPE | $\mathbf{94.82 \pm 2.91}$ |

Next, we test our method on sequence modeling in the domains of music and genomic data. Like human language, both domains have a hierarchical organization and are rule governed systems. In contrast to human language, structural rules can be quite strict and precise positional information is critical. Musical pieces contain hierarchical repeating structures (chords, phrases, motifs), where relative pitch and timing changes are more predictive than their absolute values (Huang et al., 2019). Huang et al. (2019) also note that the characteristic grammar in piano gestures relies more on relative intervals. Likewise, genomic sequences contain local patterns that rely on relative position and ordering of elements.

**Sequence modeling of symbolic music.** We train Transformer models using cross-entropy loss on MIDI-based inputs with a maximum length of 2048 from two popular music datasets, Bach-Chorales (JSB) (Boulanger-Lewandowski et al., 2012) and MAESTRO (Hawthorne et al., 2019). We closely follow the preprocessing steps from Huang et al. (2019). Table 2 reveals that PoPE achieves a decrease in negative log likelihood (NLL) compared to RoPE on both datasets.

Table 2: Best NLL on the test split for Transformer models with RoPE or PoPE positional encodings on music datasets (JSB and MAESTRO).

| Positional Enc. | JSB | MAESTRO |
|---|---|---|
| RoPE | 0.5081 | 1.501 |
| PoPE | **0.4889** | **1.486** |

**Sequence modeling of human genome.** We train a Transformer on sequences from the Human Reference Genome dataset (Dalla-Torre et al., 2025) using the standard next-token prediction loss. We follow the preprocessing and tokenization procedure from the recent state-of-the-art Nucleotide Transformer (Dalla-Torre et al., 2025) to obtain sequences with a maximum length of 1000 tokens and vocabulary size of 4107. Our PoPE-based model achieves a significant drop in NLL compared to the baseline RoPE model (Table 3).

Table 3: Best NLL on the test split for the Human Reference Genome dataset.

| Positional Enc. | Human Ref. Genome |
|---|---|
| RoPE | 4.217 |
| PoPE | **4.152** |

Although we find a benefit of incorporating PoPE into Transformers on diverse domains like music and genomic sequences, we chose these domains specifically because they appear to require the separation of position and content as well as precise positional information. It is much less clear that these properties hold true for human language. In our next set of experiments, we pretrain Transformers of varying size on OpenWebText.

**Language modeling on OpenWebText.** We test PoPE's efficacy on language modeling by training Transformers of three sizes on the OpenWebText dataset (Gokaslan & Cohen, 2019). At respective model sizes, both models use identical architecture and training parameters but differ only in the positional encoding. Across model sizes, PoPE consistently achieves lower perplexities than

Table 4: Perplexity on the validation split of OpenWebText for Transformer models with RoPE or PoPE positional encodings.

| Positional Enc. | 124M | 253M | 774M |
|---|---|---|---|
| RoPE | 21.55 | 18.88 | 15.85 |
| PoPE | **21.33** | **18.55** | **15.45** |

RoPE (Table 4). It is also interesting to note that the performance gap between RoPE and PoPE holds steady or possibly increases with model size. We run ablation experiments by training the 124M model with PoPE variants that do not use either the softplus activation, $\sigma()$, or the

learnable bias vector $\delta$ and find these components each contribute to the performance gains (refer to Appendix B for more details). Next, we move beyond perplexity to evaluate model efficacy on a standard suite of downstream tasks.

Table 5: Zero-shot performance on downstream tasks using Transformer models pretrained on OpenWebText with RoPE or PoPE positional encoding.

| Model size | Positional Enc. | LAMBADA ↑ | Blimp ↑ | CBT ↑ | HellaSwag ↑ | PIQA ↑ | ARC-E ↑ | Avg. ↑ |
|---|---|---|---|---|---|---|---|---|
| 124M | RoPE | **28.74** | 77.55 | 38.80 | **29.55** | **60.28** | 37.08 | 45.33 |
| | PoPE | 27.67 | **77.75** | **44.43** | 29.18 | 59.58 | **38.52** | **46.19** |
| 253M | RoPE | **31.38** | 79.12 | 48.51 | 31.47 | **62.24** | **39.87** | 48.76 |
| | PoPE | 30.47 | **80.01** | **49.39** | **31.82** | 61.70 | 39.32 | **48.78** |
| 774M | RoPE | 35.95 | 81.05 | 52.56 | 35.39 | 63.55 | 42.28 | 51.80 |
| | PoPE | **36.89** | **82.03** | **53.67** | **35.86** | **63.93** | **42.37** | **52.46** |

**Zero-shot performance on downstream tasks.** We evaluate the zero-shot performance of the Transformers pretrained on OpenWebText on six downstream tasks, namely: LAMBADA Paperno et al. (2016), BLiMP (Warstadt et al., 2020), Children's Book Test (CBT) (Hill et al., 2016), HellaSwag (Zellers et al., 2019), PIQA (Bisk et al., 2020), and ARC-E (Clark et al., 2018). Following Gao et al. (2024), we use the detokenized version from OpenAI for LAMBADA and evaluate the top-one accuracy on the last word (which can be multiple tokens; we use greedy decoding). For CBT and BLiMP, we measure the accuracy for each task and report the average accuracy over all tasks. Table 5 reports accuracy for three models sizes and for each of the six downstream tasks. The last column of the Table presents the mean accuracy across the tasks. For all three model sizes, PoPE-based Transformers have higher mean accuracy than RoPE-based Transformers.

**Test-time length extrapolation.** We measure the ability of PoPE to generalize to test-time sequences that are longer than those presented during training. We examine models pretrained on OpenWebText using a sequence length (context window) of 1024 tokens and assess zero-shot perplexity on much longer sequences (up to 10240 tokens) from the test split of the PG-19 dataset (Rae et al., 2020). We also compare against a strong baseline, YaRN (Peng et al., 2024), which is a state-of-the-art method used to improve the length extrapolation capabilities of RoPE. YaRN applies an interpolation scheme to the base frequencies $\theta_c$ of RoPE and finetunes the model on longer sequences. For PoPE, we simply finetune on longer sequences without interpolating the frequency components, we refer to this variant as PoPE+ft in Figure 2. Both YaRN and PoPE+ft models are finetuned on sequences of length 4096 from the OpenWebText dataset for 500 steps with a warmup of 20 steps using batch size of 64 and learning rate of 6e-5 without any decay.

In Figure 2, we see that RoPE's performance significantly degrades on longer sequences at test time. YaRN mitigates this performance degradation on sequences whose length does not exceed the size of the context window used for fine-tuning (upto 4096). However, when extending to test sequences whose length exceeds that used for fine-tuning, YaRN's performance also degrades significantly. In stark contrast, PoPE shows strong out-of-the-box length extrapolation capabilities without any fine-tuning or position interpolation techniques at test time on 10x longer sequences (Figure 2) and even outperforms specialized baselines like YaRN. Further, our fine-tuned PoPE variant, shows noticeable improvement in perplexity on longer sequences compared to PoPE, which suggests that the fine-tuning process enables the adaptation of low frequency components based on the longer sequences seen during fine-tuning. Overall, these results highlight the strengths of our simple method to overcome these critical issues of RoPE in a principled way. It is also interesting to note that RoPE's extrapolation performance degrades with model size, while PoPE's extrapolation performance remains largely stable. RoPE's failure is due to its allowance of a what-where interaction: aspects of the key and query representation can dynamically shift the position tuning of a component. These shifts become problematic, especially for the lowest frequency components, as they exert their influence only when the context window is expanded.

**Frequency usage analysis.** Barbero et al. (2025) analyze RoPE's use of features at different period lengths by plotting the mean norm over the 2D RoPE components for all layers. They observe that

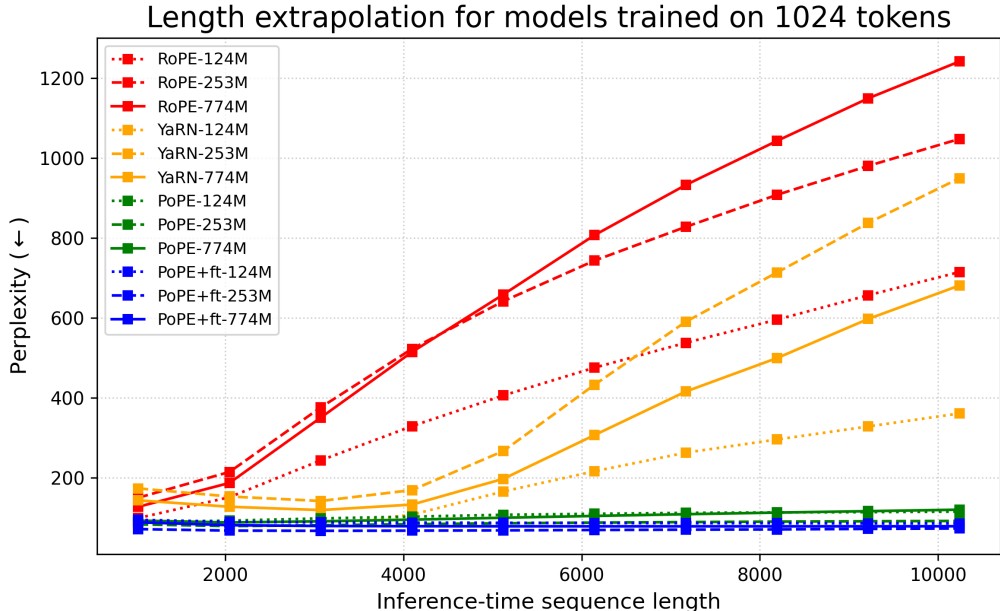

Figure 2: Length extrapolation at test-time on PG-19 dataset for different model sizes. We evaluate baselines that use RoPE (red) or YaRN (yellow) against PoPE (green) which does not apply any fine-tuning or interpolation techniques and PoPE+ft (blue) which only uses fine-tuning. Sequences at test-time are multiples of 1024 up to 10240.

the Gemma 7B model prefers to maintain low norms on the high-frequency channels to minimize interference, because the contribution of these channels to the attention dot product behaves similar to random noise. Gemma has high norms only for features in a sparse set of low frequencies. We apply their analysis on our 124M Transformer models pretrained on OpenWebText (from Table 4). In our 124M Transformer, we similarly find that RoPE produces high norms for only a sparse subset of frequency channels across all layers (Figure 3). While there are qualitative differences in Barbero et al. (2025) visualizations of Gemma 7B and our 124M RoPE baseline model, we note that the pretraining and model scales are vastly different, which might explain the different behaviors they've extracted from training data. In contrast, the PoPE Transformer assigns high norms on the high frequency features across all layers except the first (Figure 4). PoPE also shows a more distributed usage of features across the full frequency range compared to RoPE. Notice the doubling in the number of frequencies (rows of the heatmap) with PoPE (Figure 4) versus RoPE (Figure 3).

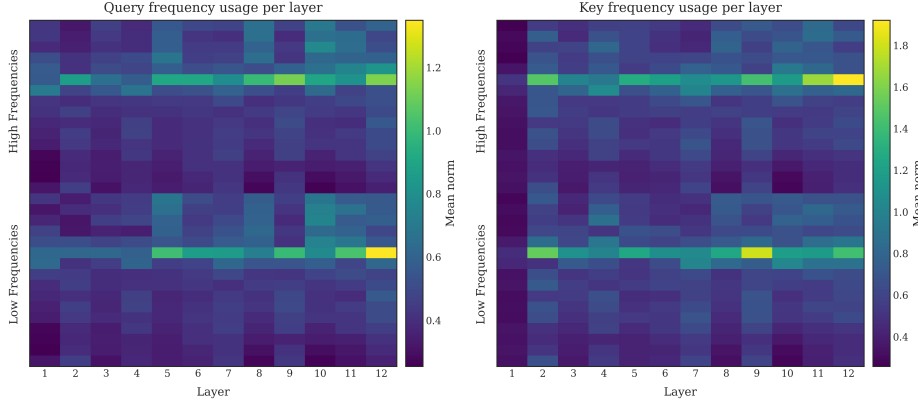

Figure 3: 2-norm plotted over 2D RoPE 'chunks' of queries (left) and keys (right) in each layer of the 124M Transformer over different RoPE frequencies. Mean over 10 different Shakespeare sonnets and 12 attention heads at each layer.

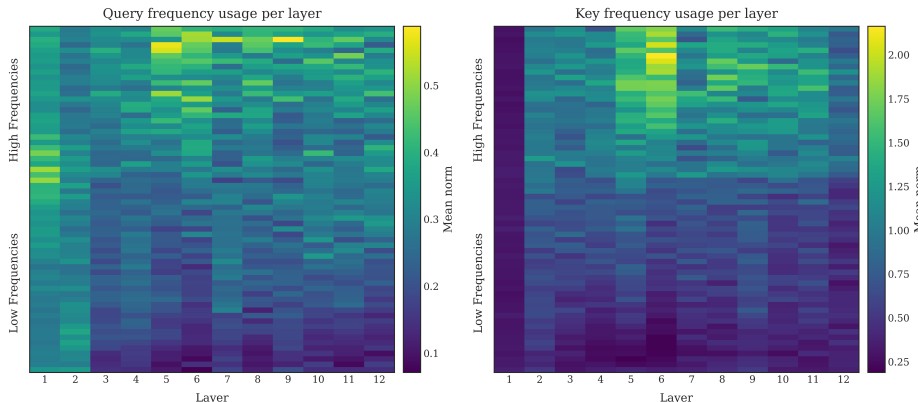

Figure 4: Magnitude of each complex-valued features of queries (left) and keys (right) in each layer of the 124M Transformer over different PoPE frequencies. Mean over 10 different Shakespeare sonnets and 12 attention heads at each layer.

## 5 RELATED WORK

**RoPE and its extensions.** The most popular positional encoding used by current LLMs (Touvron et al., 2023; Grattafiori et al., 2024; Liu et al., 2024; Team et al., 2025; Yang et al., 2025) is RoPE (Su et al., 2024). However, RoPE is known to be unable to generalize to longer sequences than it was trained on. Pretraining on long sequences is expensive; thus, the models are typically trained on relatively short context lengths, and then extended to longer context lengths during post-training. This can be done, for example, by changing the rotation frequency for each position so that the total amount of rotations for each component stays the same at the maximum context length as during pretraining (Chen et al., 2023). A more performant alternative is YaRN (Peng et al., 2024) that scales higher frequencies less than the lower ones, retaining the ability of the model to recall small relative distances precisely. Ding et al. (2024) shows that this strategy can be further improved by skipping the scaling for early token positions and searching for more optimal scaling actors using a genetic algorithm. Furthermore, they perform iterative scaling followed by tuning phases, which allows them to expand the context window to 2M tokens. Sun et al. (2023) takes a different approach: the authors add a decay factor similar to ALiBi (Press et al., 2022) and block-wise masking to limit the span of the attention during inference. Although this prevents a sudden performance drop in pure RoPE, it does not allow us to recall information from long distances. Wang et al. (2024) proposes rounding all RoPE wavelengths to integers, such that there is no increasing shift in the rotation angle after each time the given channel wraps around. This further improves the performance of YaRN, but is also effective without length extension techniques.

**Alternative positional embeddings.** The original Transformer (Vaswani et al., 2017) uses sinusoidal embeddings to encode absolute positions; which are added to the token embeddings only at the input layer. The sinusoidal positional embeddings also generally underperform w.r.t more modern methods for relative positional embeddings (Su et al., 2024) which inject this information at every layer. It has been shown that autoregressive Transformers do not require explicit encoding of positional information to operate (Schmidhuber, 1992a; Irie et al., 2019; Irie, 2025). Such Transformers tend to have better length extrapolation properties than both absolute and relative positional embeddings (Kazemnejad et al., 2023), although this comes at the expense of their in-distribution performance. In the 1990s, neural sequence models, e.g., the Neural History Compressor (Schmidhuber, 1992b), used a relative positional encoding based on the inverse time that went by since the last unexpected input. Shaw et al. (2018) introduced a relative positional embedding which uses a separate set of keys that are selected based on the distance of the key and query. Music Transformer (Huang et al., 2019) proposed a more efficient implementation of this relative positional embedding. Dai et al. (2019) propose a different variant, where instead of learning a separate key for each offset, they use a learned projection of sinusoidal "offset encodings", which are inspired by the absolute positional encodings. This approach generalizes better to longer lengths in practice. Additionally, the authors train sequentially within documents and allow attention

to the previous batch, enhancing the long-range capabilities of the models further. Wang et al. (2020) generalized this idea of "offset encodings" by using complex-valued embeddings of inputs to encode information about the content, global position and their order relationships within the sequence. T5 (Raffel et al., 2020) takes a different approach: their method groups token pairs in log-sized buckets based on their distance, and it adds a learned per-bucket bias to the attention logit, allowing it to decrease the importance of far-away context. ALiBi Press et al. (2022) takes inspiration from this, adding learned scores to the attention matrix that decay with relative distance. Geometric (Csordás et al., 2022) or stick-breaking attention (Tan et al., 2025) take a radically different approach: it replaces the softmax activation function with a stick-breaking process that gives priority to good matches that are nearby without explicitly encoding positions or offsets. The authors claim superior length generalization.

# 6 CONCLUSION

We proposed a new relative positional encoding technique called PoPE whose query-key attention scores are based on a computation that decouples the match based on content and the match based on position. In contrast, RoPE confounds the 'what' and 'where' information between keys and queries which leads to difficulties in learning to match based solely on 'what' or 'where', as we highlight via a diagnostic task we introduce called Indirect Indexing. A RoPE-based Transformer struggles to solve this task whereas a PoPE-based Transformer learns this task easily. On autoregressive sequence modeling in music, genomic, and natural language domains, Transformers using PoPE as the positional encoding scheme outperform baselines using RoPE with respect to training loss (perplexity) and downstream task performance. On language modeling, these gains persist across model scale, from 124M to 774M parameters. Crucially, PoPE shows strong zero-shot length extrapolation capabilities, whereas RoPE's performance degrades significantly on longer sequences at test time and requires fine tuning or position interpolation methods.

# 7 ETHICS STATEMENT

We consider our work to be fundamental research on sequence modeling with Transformers with no direct societal implications. However, our work which develops a novel relative positional encoding scheme that improves sequence modeling capabilities in Transformer models could potentially lead to benefits as well as unforeseen harms and risks from dual-use applications. By improving sequence modeling capabilities across domains such as music, genomics and natural language, this work could potentially enhance the performance of Foundation Models used for downstream applications such as protein folding prediction, drug discovery, music composition and conversational assistance. The enhanced length extrapolation capabilities of our method are particularly valuable for real-world deployment of large language models, especially variants that use chain-of-thought as they process much longer sequences at test-time, potentially reducing the amount of fine tuning on longer sequences and the associated computational costs. However, as with any advancement in the capabilities of Foundation Models, it could potentially be misused for generating harmful content at scale or creating more sophisticated deepfakes in text, audio, images etc.

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

# A EXPERIMENTAL DETAILS

Following sections provide details on datasets, preprocessing, models and training configurations.

## A.1 DATASETS

**Indirect Indexing.** This diagnostic task (Indirect Idx.) requires the model to locate a target character in a variable-length source string that is at a certain relative distance (left or right) from a source character. It tests for the structured manipulation (pointer arithmetic operations) of the content and positional information of tokens by a sequence processing architecture. The dataset for this task is constructed by procedurally generating examples of source strings, source character and relative shifts. We generate source strings of length between 20 and 40 characters from the set of uppercase [A-Z] and lowercase [a-z] letters by uniform sampling without replacement. Then, we randomly pick a source character given a randomly sampled source string. Next, we uniformly sample shifts in the range [-15, +15] (where - and + indicate left and right shifts respectively) and consequently obtain our target character. We generate a train/validation/test splits of size 1M/10k/10k respectively. We use character-level tokenization, i.e. all uppercase and lowercase letters, individual digits, the delimiter symbol, plus and minus signs are separate tokens. The format of each examples is: <source string>, <source character>, <shift>, <target character> and ',' as a delimiter and the model is given the entire sequence except the target character. Below are a few samples from the dataset for reference.

```
TzbkWoKDyscBepYvfwxEVQtgPa, c, -8, b
NZTUIGWkXFrhCJDzscat, N, +4, I
RBEvOPgtaGDnjhbJCLScruZpMNsyWfQxXFAzUT, x, +2, F
```

**OpenWebText.** This dataset is an open source effort to reproduce OpenAI's WebText dataset (Gokaslan & Cohen, 2019) which was used to train GPT-2 (Radford et al., 2019). The training and validation splits roughly contain 9B and 4M tokens respectively and maximum sequence length of 1024 for pretraining. We use the GPT-2 tokenizer with a vocabulary size of 50257.

**Bach-Chorales.** This dataset (JSB) consists of 4-part scored choral music, which are represented as a matrix with rows corresponding to voices and columns to time discretized to 16th notes. The entries in this 2D matrix are integers that denote the pitch being played. We serialize this matrix in raster-scan fashion by first going down the rows and then moving right through the columns as in prior work (Huang et al., 2019). We use the variant of the dataset with 16th note temporal "quantizations" where silence is represented by a pitch of -1 rather than NaN, available in JSON file format [2]. We use a maximum sequence length of 2048 for training with 229/76/77 sequences present in the train/validation/test sets. We use a vocabulary size of 90 which includes the MIDI notes, silence and padding tokens.

**MAESTRO.** The dataset contains about 200 hours of paired audio and MIDI recordings from ten years of International Piano-e-Competition. The MIDI data includes key strike velocities and sustain/sostenuto/una corda pedal positions. We use version v3.0.0 of this dataset in MIDI format [3] for our experiments. We apply data augmentation by using pitch transpositions sampled uniformly from $\{-3, -2, -1, 0, +1, +2, +3\}$ similar to prior work (Huang et al., 2019), divide it into sequences with a maximum length of 2048 and use a 90/5/5 percent split for train/validation/test sets. We use the REMI tokenizer (Huang & Yang, 2020) with EOS, BOS, MASK and PAD tokens leading to a total vocabulary size of 328.

**Human Reference Genome.** The human reference genome (HRG) dataset was constructed by considering all autosomal and sex chromosomes sequences from reference assembly GRCh38/hg38 [4] and reached a total of 3.2 billion nucleotides. We follow the preprocessing and tokenization procedures from the recent state-of-the-art model for genomic sequence modeling, the Nucleotide Transformer (Dalla-Torre et al., 2025).

---

[2] https://github.com/czhuang/JSB-Chorales-dataset
[3] https://magenta.withgoogle.com/datasets/maestro
[4] https://huggingface.co/datasets/InstaDeepAI/human_reference_genome

**PG-19.** This dataset includes a set of books extracted from the Project Gutenberg books library, that were published before 1919. It is a popular dataset first introduced by (Rae et al., 2020) to benchmark long-range language models. In this work, we use it to evaluate the test-time length extrapolation capabilities of Transformer models that use different relative positional encoding schemes. We use the test split of the dataset containing 100 books or roughly 7M tokens. [5]

## A.2 MODELS

Table 6 contains the Transformer model configuration used on each dataset.

Table 6: Transformer model configurations for the different datasets. For the OpenWebText language modeling dataset we train 3 model sizes 124M/253M/774M and the hyperparams for each of these model sizes as a triple in column 2.

| Hyperparameter | Indirect Idx. | OpenWebText | JSB | MAESTRO | HRG |
|---|---|---|---|---|---|
| Embedding size | 512 | 768/1024/1280 | 256 | 384 | 1024 |
| Num. heads | 8 | 12/16/20 | 8 | 8 | 16 |
| Num. layers | 8 | 12/16/36 | 6 | 6 | 16 |
| Norm. type | RMSNorm | RMSNorm | RMSNorm | RMSNorm | RMSNorm |
| Base wavelength ($\theta$) | 10,000 | 10,000 | 10,000 | 10,000 | 10,000 |
| Init. range for $\delta$ | $2\pi$ | 0/0/0 | $2\pi$ | $2\pi$ | $2\pi$ |
| Dropout | 0.0 | 0.0/0.0/0.0 | 0.2 | 0.1 | 0.1 |

## A.3 TRAINING DETAILS

Table 7 contains the Transformer model hyperparameters for all the datasets.

Table 7: Hyperparameter configurations for training on different datasets. For OpenWebText dataset we train Transformer models at sizes 124M/253M/774M using identical hyperparameters.

| Hyperparameter | Indirect Idx. | OpenWebText | JSB | MAESTRO | HRG |
|---|---|---|---|---|---|
| Batch size | 64 | 64 | 4 | 16 | 64 |
| Sequence length | 40 | 1024 | 2048 | 2048 | 1000 |
| Learning rate | 2e-4 | 6e-4 | 6e-4 | 6e-4 | 2.5e-4 |
| Min. learning rate | 2e-5 | 6e-5 | 6e-5 | 6e-5 | 2.5e-5 |
| Weight decay | 0.01 | 0.01 | 0.01 | 0.01 | 0.01 |
| Grad. clipping | 1.0 | 1.0 | 1.0 | 1.0 | 1.0 |
| $\beta_2$ for AdamW | 0.99 | 0.95 | 0.99 | 0.99 | 0.999 |
| Max. iters | 100,000 | 100,000 | 3000 | 60,000 | 100,000 |
| Decay iters | 100,000 | 100,000 | 3000 | 60,000 | 100,000 |
| Warmup iters | 4000 | 1000 | 10 | 500 | 4000 |

---

[5] https://huggingface.co/datasets/emozilla/pg19-test

## B ADDITIONAL RESULTS

To test the efficacy of key design choices, we run ablations by training 124M Transformer models on OpenWebText with identical configurations, comparing our full PoPE method against variants that remove either the softplus activation $\sigma()$ or the bias vector $\delta$ that tunes the optimal relative offset for each channel $c$. From Table 8 we see that by removing each of these components the performance of our PoPE variants worsens with increases in perplexity scores. The full PoPE model outperforms these ablated variants and indicates the efficacy of these design choices.

Table 8: Validation set perplexity scores for the 124M Transformer model with ablated versions of PoPE pretrained on OpenWebText.

| Positional Encoding | 124M |
|---|---|
| PoPE without $\sigma()$ | 21.57 |
| PoPE with ReLU for $\sigma()$ | 21.55 |
| PoPE without $\delta$ | 21.42 |
| Full PoPE | **21.33** |

## C ADDITIONAL VISUALIZATIONS

**Frequency usage analysis.** We apply the frequency usage analysis for the 253M-sized Transformer model pretrained on OpenWebText (from Table 4). From Figure 5 and Figure 6, we see a similar pattern emerge as before, i.e. RoPE tends to use a sparse set of frequencies indicated by the high norms and does not use the highest frequencies. Whereas PoPE uses a broader set of frequencies most notably the highest ones.

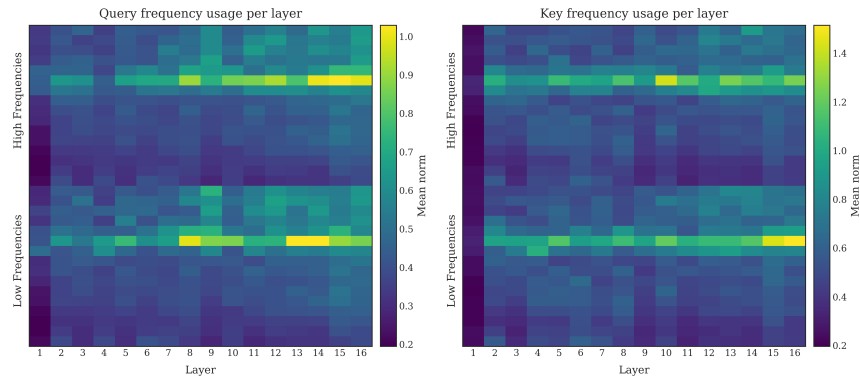

Figure 5: 2-norm plotted over 2D RoPE components of queries (left) and keys (right) in each layer of the 253M Transformer over different RoPE frequencies. Mean over 10 different Shakespeare sonnets and 16 attention heads at each layer.

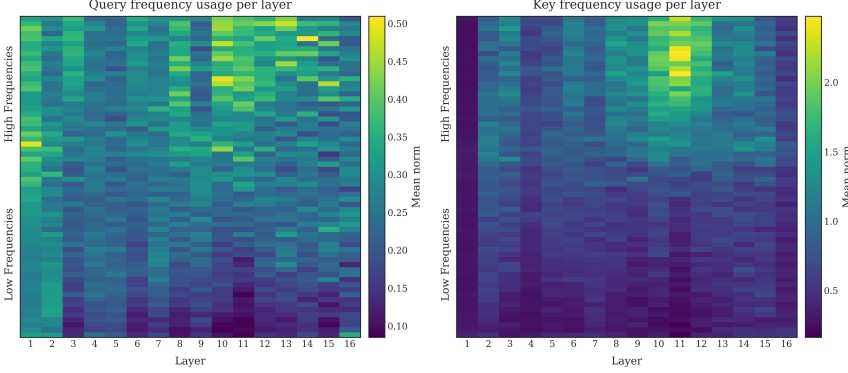

Figure 6: Magnitude of each complex-valued features of queries (left) and keys (right) in each layer of the 253M Transformer over different PoPE frequencies. Mean over 10 different Shakespeare sonnets and 16 attention heads at each layer.

**Visualization of the learnable bias.** Below we visualize the learned biases $\delta_c$ from the pretrained PoPE models on OpenWebText from Table 4. Note that the biases were initialized as zeros and are restricted to lie in the range $-2\pi$ to $0$. The one consistent pattern across both model sizes is that the bias values ($\delta_c$) tend to deviate away from zero only for the higher frequency components, while the low frequency components remain at their initial zero values.

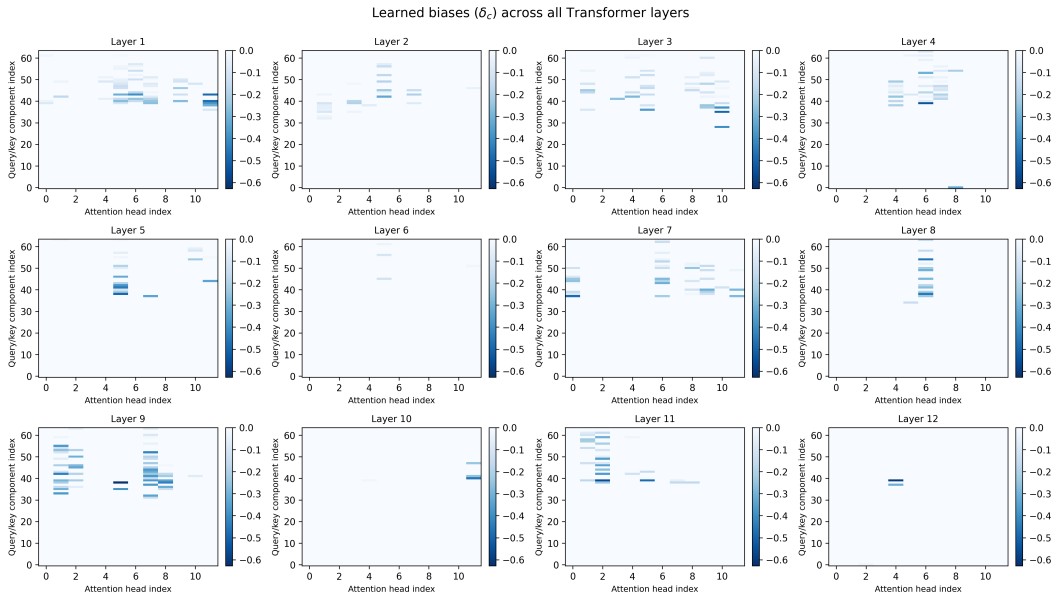

Figure 7: Visualization of the learned biases for the 124M pretrained PoPE model across all layers.

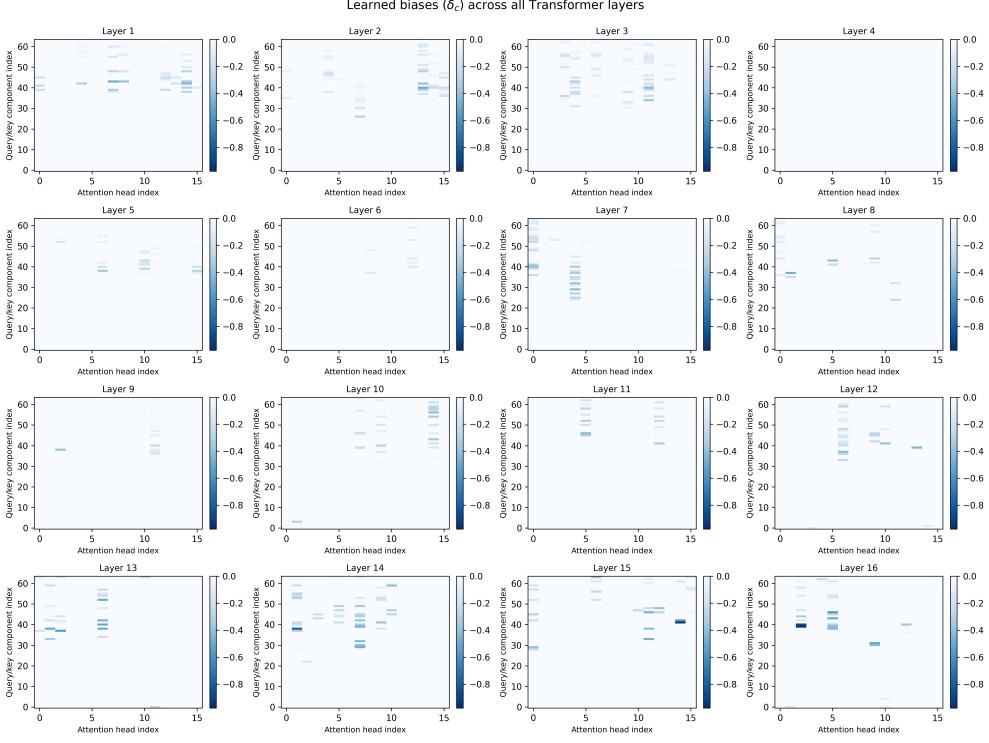

Figure 8: Visualization of the learned biases for the 253M pretrained PoPE model across all layers.

