# OpenReview forum: "Decoupling The "What" and "Where" With Polar Coordinate Positional Embedding"
_ICLR.cc/2026/Conference — Submitted to ICLR 2026_

### Official Review · Reviewer_VCmS · 2025-10-22

**Soundness:** 3
**Presentation:** 4
**Contribution:** 3
**Rating:** 6
**Confidence:** 4

**Summary:**

The authors introduce Polar Coordinate Position Embeddings (PoPE), an improvement to RoPE which uses polar coordinates to encode positional information. The primary benefit is that by disentangling content and position in RoPE through the use of angle and magnitude information, there is better mixing of information that can be performed by the model in a context-specific manner. This is demonstrated through superior performance in various sequence modeling tasks and zero-shot length extrapolation.

**Strengths:**

- The method is motivated mathematically and intuitive to follow.
- The method is simple and not architecture dependent (aside from being used in Transformers), thus I am under the impression it can be integrated with any existing method on this front.
- The experiments are comprehensive and demonstrate the benefits of PoPE in relation to RoPE.

**Weaknesses:**

- While the method does show improvements relative to RoPE in the context of the models being studied, some standard benchmarks could be included (ex. LongBench, RULER, etc.) to better show some of the long-context scaling behaviour beyond just perplexity. Additionally, results after instruction-tuning the model would be appreciated to better highlight the benefits of the method.
- One small consideration is that given the additional parameters that are stored for RoPE, this could affect cache optimization at inference time. I would expect this to potentially lead to come limitations on this front, and this could also lead to some problems with current cache optimization techniques or inference-time scaling techniques (ex. attention sinks or meta-tokens).

**Questions:**

See above.

---

> ### Author Response · Authors · 2025-11-24
> **Rebuttal Response**
>
> Please note that our models are trained on a very modest academic computational budget, thus they are very small. So, we do not see any benefit to instruction tuning in our case with these very small models on Q&A datasets. Further, our method is a change in the positional encoding and the focus of our paper has been on the advantages it offers in pretraining, we believe our experiments support the main claims in our work.
>
> However, we have added a comparison to YaRN for the length generalization experiments (see updated Figure 2), and fine-tuning with a long context of 4096 with our method where we do not apply any interpolation of the rotation frequencies. We show that our methods (PoPE and PoPE+ft) outperforms YaRN significantly, and unlike YaRN, it generalizes strongly beyond the fine-tuning length. Most strikingly, is that the PoPE model which does not use any fine-tuning or interpolation of frequencies still strongly outperforms specialized baselines like YaRN, highlighting the strength of our simple approach.
>
> With regards to the memory overhead, in our current naive implementation, this is true. However, note that, as discussed in the paper in L189-190, it is possible to write a Flash Attention-style kernel that performs the rotation within the kernel itself, after loading the data in shared memory. In this case, the key and query sizes and the required memory bandwidth exactly match those of RoPE as only the magnitude components need to be cached. The only remaining cost is an extra matrix multiplication, but because attention is typically memory-bound, that should only add a minimal overhead.

---

> > ### Comment · Reviewer_VCmS · 2025-11-27
> >
> > I thank the authors for their response.
> >
> > Ultimately, I think these larger models as well as a working implementation that resolves these issues could be beneficial, as well as possible further experiments on models such as linear attention where the position embedding is quite important as well.
> >
> > Nevertheless, I maintain my positive assessment of this work.

---

### Official Review · Reviewer_LYyY · 2025-10-27

**Soundness:** 2
**Presentation:** 3
**Contribution:** 2
**Rating:** 4
**Confidence:** 4

**Summary:**

This paper proposes Polar Coordinate Positional Embedding (PoPE), a modification of the widely used Rotary Position Embedding (RoPE) for Transformer architectures. The authors argue that RoPE entangles content (“what”) and position (“where”) information in query–key attention, which may hinder performance on tasks that require independent reasoning about these two aspects.

PoPE reformulates RoPE in polar coordinates, where the magnitudes (content) and phases (position) are explicitly separated. The magnitudes of queries and keys are made positive via a softplus transformation, and the position is represented purely by the phase terms. This eliminates the cross-term between the query/key phases that causes what–where coupling in RoPE.

Empirically, PoPE shows strong improvements across multiple domains: 1. Indirect Indexing diagnostic task: 95% vs 11% accuracy (RoPE) 2. Sequence modeling: lower NLL on music (JSB, MAESTRO) and genomic (HRG) datasets; 3. Language modeling: reduced perplexity on OpenWebText across 124M–774M parameter scales 4. Downstream and extrapolation tasks: better zero-shot accuracy and robust long-context generalization without interpolation.

**Strengths:**

1. The paper provides a geometric explanation of RoPE’s limitation, mixing content and position, and introduces PoPE as a principled disentanglement.
2. The polar-coordinate derivation clearly isolates the “interaction term” $\phi_k-\phi_q$ and justifies its removal to achieve independence between what/where components.
3. Experiments span diagnostic synthetic tasks, structured symbolic domains (music, genomics), and large-scale language modeling, showing consistent, domain-general gains.
4. The length extrapolation results (up to 10× longer sequences) are compelling and address a long-standing weakness of RoPE-based models.
5. PoPE can be dropped into any Transformer with almost no code change and negligible computational overhead (only one extra multiplication).

**Weaknesses:**

1. The “disentanglement” argument is mostly qualitative. There is no formal analysis or proof that the removed phase interaction necessarily leads to improved generalization.

2. Softplus is heuristic. While used for ensuring positive magnitudes, its necessity and effect are not theoretically grounded beyond empirical ablation.

3. It would be useful to see comparisons to YaRN, ALiBi, or other strong length-extrapolatable positional encodings, not just RoPE.

4. While conceptually appealing, it remains unclear how the model behaviorally separates content vs. position in downstream reasoning.

**Questions:**

1. Can you formalize why removing the $\phi_k-\phi_q$ interaction improves expressivity or inductive bias, beyond the empirical results?
2. Did you test other non-negative mappings (e.g., ReLU, square, exponential)? How sensitive are results to the choice of activation?
3. Could this reintroduce a form of coupling or shift-dependent bias that undermines the core disentanglement principle?
When $s \approx t$, the query and key correspond to nearby positions in the sequence, and one would expect
$\cos((t - s)\theta_c) \approx 1$ in Eq.(5).
However, introducing a learnable bias term $\delta_c$ could force
$\cos((t - s)\theta_c + \delta_c) \neq 1$,
which appears counterintuitive and may distort locality or positional consistency.

---

> ### Author Response · Authors · 2025-11-24
> **Rebuttal Response**
>
> Regarding the comparison of PoPE to other strong length extrapolable positional encodings, we have updated Figure 2 which now includes a comparison to YaRN as well. For the YaRN baseline, we took the pretrained RoPE model (pretraining context length of 1024) and applied YaRN enhancements (linear interpolation of base frequencies Eq. 18-20 in YaRN paper) and fine-tuned it on context length of 4096 on OpenWebtext dataset. We also fine-tuned our pretrained PoPE model (“PoPE+ft” in updated Figure 2) on the longer sequences (context length of 4096) from the OpenWebText dataset. Please note that the fine-tuning process for PoPE does not apply any linear interpolation or modify the base frequencies in any way but simply fine-tunes the pretrained model on longer sequences. For evaluation, as before we measure the perplexities on sequences of varying lengths from the PG-19 test split. In the updated Figure 2, we can see that both PoPE and PoPE+ft strongly outperform RoPE and YaRN baselines.
>
> > Can you formalize why removing the $\phi_k - \phi_q$ interaction improves expressivity or inductive bias, beyond the empirical results?
>
> Strictly speaking, PoPE, which constrains $\phi_k-\phi_q$ to be zero, reduces expressivity: it is a special case of RoPE. It therefore embodies a stronger inductive bias than RoPE. It requires solutions in which the content match and the positional match have a multiplicative (conjunctive) effect on the overall match score.  It is in this sense that the method disentangles.  (Disentangling may have been a poor choice of wording, but it comes from the literature on factorial learning where disentangled representations have the probability that the joint probability of two features is the product of the marginals.  Here, the effect on the attention score is the product of the embedding match and the position match.)
>
> We experimented with other choices of activations for the non-negativity constraint such as ReLU and found that Softplus worked the best. We added the results of the ReLU variant to Table 8 in Appendix B.
>
> > Could this reintroduce a form of coupling or shift-dependent bias that undermines the core disentanglement principle?
>
> Please note that the attention score computation in Eq.6 for PoPE with the cosine term ensures that matches based on position are a function of the relative separation between the query and key (t-s) and not their absolute positions in the sequence. Simply, we want a particular query and key component with small separation, i.e. small value for (t-s), to be upweighted more than those with a large separation, i.e. high value for (t-s), which is satisfied by PoPE. Without the learnable bias $\delta_c$, the highest attention score for all components will always be at t=s, i.e. the query attends strongest to the key at the same position. The learnable bias $\delta_c$ simply allows the model to tune the absolute position index at which different components c get the highest attention score. It does not change the fact that query-key pairs are down weighted in proportion to their relative separation (t-s) in the sequence. Further, the disentanglement we have advocated for is between the matches based on content and position. In Eq. 6, matches based on content are from the dot-product of the $\mu$ terms while matches based on relative position are from the cosine term. The learnable bias does not provide the model a way to apply rotations to query/key vectors in a content-dependent manner. Therefore it does not undermine the core disentanglement principle.

---

### Official Review · Reviewer_Y9wq · 2025-10-31

**Soundness:** 2
**Presentation:** 2
**Contribution:** 1
**Rating:** 2
**Confidence:** 5

**Summary:**

In this paper, the authors rewrite RoPE positional encoding in polar coordinates and discover that the phase of the cosine function in RoPE is related not only to relative position ("where," i.e., $(s - t)\theta_c$ in Equation (2)), but also to the query/key themselves ("what," i.e., $\phi_{k_{sc}} - \phi_{q_{tc}}$ in Equation (2)). The authors argue that the coupling effect of 'what' and 'where' on the phase of the cosine function may be detrimental to model performance. Therefore, they propose PoPE positional encoding, which removes the 'what' term from the cosine function's phase, so that the phase depends solely on relative positional relationships. In their experimental setup, the authors compare RoPE and PoPE on tasks such as Indirect Indexing, synthetic music sequence modeling, human genome sequence modeling, language modeling and downstream evaluation, as well as length extrapolation during testing.

**Strengths:**

(1) PoPE shows an improvement over RoPE on the Indirect Indexing synthetic task. This task involves string sequences where, given a source character and an offset relative to that character, the model needs to retrieve the target character obtained by applying the offset to the source. Experimental results demonstrate that, compared to RoPE, PoPE—whose cosine function phase depends only on relative positional relationships—enables more accurate localization for this type of task.

(2) PoPE demonstrates better length extrapolation ability than RoPE during testing. According to Figure 2 in the paper, when models are trained on sequences of length 1024 and then evaluated on sequences up to 10240 in length, PoPE’s language modeling perplexity loss remains nearly unchanged. In contrast, RoPE shows a noticeable increase in perplexity, highlighting PoPE’s advantage in handling longer sequences against RoPE.

**Weaknesses:**

The paper lacks citations and discussion of works with very similar formulations. This work shares an almost identical formulation with [1] and complex version of [2], yet it neither cites nor discusses the relationship between them. Additionally, the improvements appear marginal, and there is no efficiency/speed comparison provided.

Citations:
1. https://arxiv.org/abs/2202.08791
2. https://arxiv.org/abs/2307.09270

**Questions:**

See the weaknesses part.

---

> ### Author Response · Authors · 2025-11-24
> **Rebuttal Response**
>
> We appreciate the reviewer citing the two interesting earlier papers which share related themes with our work. However, the reviewer describes these papers as having “very similar formulations.” We strongly reject the notion that these papers scoop or otherwise overshadow our work, and we argue below that the similarities are superficial.
>
> The paper entitled “Linearized relative positional encoding” was accepted for publication in TMLR and the editor described the work as proposing *“...a formalism for positional embeddings in the case of linear transformers… The formalism encompasses the ROPE embeddings as a special case.”* We highly value papers like this one that lay out a space of algorithms that allow one to see the relationship among algorithms, and our PoPE algorithm lies in this space (except for the linearity constraint). However, laying out a broad space of algorithms does not obviate the need to explore specific algorithms within the space. There is no mention of a PoPE-like scheme within the article. Indeed, the authors state  that they *“put together a variety of existing linear relative positional encoding approaches under a canonical form”*. We’re very happy to acknowledge that PoPE lies in this family (again, barring the linearity constraint), but to dismiss our work simply because this framework exists is unjustified.
>
> The paper titled, “cosFormer: Rethinking softmax in attention”, is—like the first—focused on linear transformers and the efficiency that can be realized by eliminating the softmax operator. We note that the issue of linearity vs. softmax is orthogonal to the position coding in principle. However, in the authors’ formulation of the cosFormer, they do include an equation that—superficially—is similar to an equation in our work.  The key equation, Equation 10, describes a similarity score that is the product of a key-query match and a cosine function of the positional difference. Our work also hinges on an equation that can be described similarly. However, there is a fundamental difference in the role of the cosine function.  In the cosFormer, the function is $\text{QKMatch} * \cos(M * (i-j))$ where i and j are the key and query token indices and M is a constant. In our work, the function is $\sum_h k_h q_h * \cos(M_h * (i-j))$. Note that in our work, the position difference has a different effect by key and query component (indexed by h), whereas in the cosFormer, the cosine term serves to bias the selection of keys that are at predefined distances from the query. Essentially, in the cosFormer, the model prefers to match entire keys to entire queries when they are an optimal distance apart (determined by a constant M), whereas in PoPE, the model matches individual components based on a component-specific ideal separation. PoPE, like RoPE, breaks the key and query into components, whereas cosFormer considers the global key-query match. In terms of the math, both RoPE and PoPE are framed in terms of complex arithmetic operations, whereas the cosFormer is not.

---

### Official Review · Reviewer_TL8Y · 2025-11-03

**Soundness:** 2
**Presentation:** 2
**Contribution:** 2
**Rating:** 4
**Confidence:** 3

**Summary:**

This paper presents an analysis of the popular Rotary Position Embedding (RoPE), arguing that it entangles content ("what") and position ("where") information within the attention mechanism. The authors hypothesize that this entanglement impairs model performance, especially on tasks requiring independent matching of these two factors. To address this, they propose Polar Coordinate Positional Embedding (PoPE), a novel method that decouples content and position by re-framing queries and keys in polar coordinates, where content is mapped to magnitude and position to phase.

**Strengths:**

The paper offers a clear and insightful diagnosis of a fundamental limitation in RoPE—the entanglement of content and position. The proposed PoPE method is theoretically clean, directly addressing the identified issue with scientific taste. The central claims are backed by strong experimental evidence across diverse domains (synthetic, language, music, genomics) and model scales. In particular, the method's ability to achieve robust zero-shot length extrapolation without special tricks is highly compelling and demonstrates significant value.

**Weaknesses:**

1. In its current implementation, PoPE doubles the K/V cache memory footprint, which is a serious practical barrier for training and deploying large-scale models.
2. The length extrapolation evaluation lacks a direct comparison to RoPE augmented with established extension techniques like YaRN or PI, making it difficult to assess its relative advantage over existing SOTA solutions.
3. The paper does not deeply investigate the role or the learned patterns of the `δc` bias term, despite showing it is critical for performance.
4. On standard language modeling benchmarks like OpenWebText, the performance improvements from PoPE, while consistent, are relatively modest in absolute terms.
5. The finding that different `δc` initialization strategies benefit in-distribution vs. extrapolation performance suggests the method's optimal configuration may be task-dependent, potentially complicating its use.

**Questions:**

1. Could you estimate the potential throughput (TFLOP/s) overhead of a memory-optimized PoPE kernel compared to a standard Flash Attention implementation?
2. How does PoPE's performance on the PG-19 length extrapolation task compare against a RoPE baseline that has been enhanced with YaRN?
3. Could you provide a visualization of the learned `δc` values to reveal if any structural patterns emerge across layers or frequencies?
4. Is RoPE's failure on the "Indirect Indexing" task highly sensitive to the choice of the `base` frequency `θ`? Would changing `θ` mitigate this failure?
5. The softplus function is crucial for decoupling. Did you experiment with other non-negativity constraints (e.g., ReLU), and how did they perform? This would help clarify its exact role.
6. The evaluation of long-context capabilities is narrow, resting on perplexity on a single benchmark (PG-19). This assesses modeling fluency but fails to test functional abilities like long-range information retrieval or QA, which are crucial for substantiating the paper's primary contribution.

---

> ### Author Response · Authors · 2025-11-24
> **Rebuttal Response**
>
> > In its current implementation, PoPE doubles the K/V cache memory footprint, which is a serious practical barrier for training and deploying large-scale models.
>
> With regards to the memory overhead, in our current naive implementation, this is true. However, note that, as discussed in the paper in L189-190, it is possible to write a Flash Attention-style kernel that performs the rotation within the kernel itself, after loading the data in shared memory. In this case, the key and query sizes and the required memory bandwidth exactly match those of RoPE as only the magnitude components need to be cached. The only remaining cost is an extra matrix multiplication, but because attention is typically memory-bound, that should only add a minimal overhead.
>
> Regarding the comparison of PoPE to YaRN, we have updated Figure 2 and added a comparison to YaRN as well. For the YaRN baseline, we took the pretrained RoPE model (pretraining context length of 1024) and applied YaRN enhancements (linear interpolation of frequency components, i.e. Eq. 18-20 in the YaRN paper) and fine-tuning on longer sequences with context length of 4096 on OpenWebtext dataset. We also fine-tuned our pretrained PoPE model (“PoPE+ft” in updated Figure 2) but did not apply any interpolation on the frequencies on the longer sequences with a context length of 4096 from the OpenWebText dataset. For evaluation, as before we measure the perplexities on sequences of varying lengths from the PG-19 test split up to 10240. In the updated Figure 2, we can see that both PoPE and PoPE+ft strongly outperform RoPE and specialized baselines like YaRN for extending context length.
>
> We have added visualizations of the learned biases in Figures 7 and 8 in Appendix C for the 124M and 253M pretrained models on OpenWebText from Table 4. The one consistent pattern across both model sizes is that the bias values tend to deviate away from zero only for the higher frequency components, while the low frequency components remain at their initial zero values.
>
> > Is RoPE's failure on the "Indirect Indexing" task highly sensitive to the choice of the base frequency θ? Would changing θ mitigate this failure?
>
> We ran an additional sweep over base frequency $\theta$ = [200, 500, 1000, 2000] for RoPE on the Indirect Indexing task and the results are shown below. Note that $\theta$ was originally set to 10000 for both RoPE and PoPE for the results shown in Table 1.
>
> |$\theta$ | Task Acc.        |
> |----------|------------------|
> | 200       | 33.08 +- 16.36 |
> | 500       | 39.69 +- 12.70 |
> | 1000     | 17.16 +- 5.96   |
> | 2000     | 15.26 +- 4.36   |
>
> While there is some variance in the task accuracies for the RoPE model when tuning the base frequency hyperparameter, it still struggles to solve the task compared to PoPE which works very well with an off-the-shelf choice of 10000 for the base frequency $\theta$ on this task highlighting its robustness.
>
> > The softplus function is crucial for decoupling. Did you experiment with other non-negativity constraints (e.g., ReLU), and how did they perform? This would help clarify its exact role.
>
> We experimented with other choices of activations for the non-negativity constraint such as ReLU and found that Softplus worked the best. We added the results of the ReLU variant to Table 8 in Appendix B.

---

### Meta-Review · Area_Chair_1vSo · 2026-01-13

**Summary:**

The reviewers broadly agree that the paper is well-motivated and technically sound, with a clear diagnosis of the entanglement between content (“what”) and positional (“where”) information in RoPE and a conceptually simple modification (PoPE). Multiple reviewers highlight the strong empirical results on diagnostic tasks and the compelling zero-shot length extrapolation behavior compared to vanilla RoPE (e.g., Reviewer TL8Y: “clear and insightful diagnosis”; Reviewer LYyY: “compelling … address a long-standing weakness of RoPE-based models”).

However, several substantial concerns remain unresolved, which collectively inform my recommendation to reject:
- **Practical efficiency and deployability are a major concern**: the current PoPE implementation doubles the K/V cache memory footprint, which Reviewer TL8Y describes as a “serious practical barrier for training and deploying large-scale models.” Although the authors argue that a FlashAttention-style kernel could mitigate this overhead, no concrete implementation, runtime measurements, or memory benchmarks are provided, leaving this concern largely speculative.

- **Novelty and related-work positioning remain contentious**. Reviewer Y9wq, with high confidence (confidence score 5), argues that the formulation is “almost identical” to prior work and that the paper “lacks citations and discussion of works with very similar formulations.” While the authors strongly dispute this claim, the rebuttal does not fully resolve the concern by clearly situating PoPE relative to these works in a neutral and comprehensive comparison.

- **While the authors added a comparison to YaRN and additional ablations, the evaluation of long-context capabilities remains narrow, focusing primarily on perplexity on PG-19**. Reviewers (e.g., TL8Y, VCmS) explicitly note the lack of functional long-context evaluations or broader efficiency comparisons, which weakens the practical impact of the claims.

Taken together, despite the paper’s conceptual clarity and promising results, the unresolved concerns regarding efficiency, positioning with respect to prior work, and limited evaluation scope prevent a confident recommendation for acceptance.

**Reviewer Concerns:**

## Concerns discussed/addressed (partially) by the rebuttal

- **Comparison to YaRN for length extrapolation**:
Reviewers TL8Y and LYyY requested a direct comparison to strong RoPE-based extrapolation baselines such as YaRN. The authors addressed this by adding YaRN results in the updated Figure 2 and describing the fine-tuning and evaluation protocol (“we have updated Figure 2 and added a comparison to YaRN as well”). This partially addresses the concern that PoPE was only compared against vanilla RoPE.

- **Sensitivity to RoPE base frequency θ**:
Reviewer TL8Y questioned whether RoPE’s failure on the Indirect Indexing task was sensitive to θ. The authors responded with an explicit sweep over θ ∈ {200, 500, 1000, 2000}, showing that RoPE remains unstable and significantly worse than PoPE. This is a clear and adequate response to that specific concern.

- **Ablation of the non-negativity constraint (Softplus vs. ReLU)**:
Reviewers TL8Y and LYyY asked whether Softplus was essential. The authors report additional experiments with ReLU and include results in Appendix B, concluding that Softplus performs best. While largely empirical, this sufficiently addresses the narrow ablation request.

- **Visualization of the learned $\delta$c bias**:
Reviewer TL8Y requested insight into the learned $\delta$c term. The authors added visualizations (Appendix C) and describe consistent patterns (biases deviating from zero mainly at high frequencies), which addresses the request at a descriptive level.

## Concerns that remain outstanding

- **Efficiency and memory overhead of PoPE**:
Reviewer TL8Y emphasizes that PoPE “doubles the K/V cache memory footprint,” calling it a “serious practical barrier.” The rebuttal argues that a FlashAttention-style kernel could avoid this overhead, but provides no implementation, benchmarks, or empirical validation. As such, the concern about real-world efficiency and deployability remains unresolved.

- **Novelty and relation to prior work**:
Reviewer Y9wq argues that the method has an “almost identical formulation” to prior work and criticizes the lack of citations and discussion. Although the authors strongly reject this interpretation and provide qualitative distinctions, they do not clearly incorporate or reconcile these works in a systematic comparison. Despite the referred works relate to Linear Transformer topic, it should be done to carefully discuss the additional value brought by this submission based on these existing formulations.

- **Limited scope of long-context evaluation**:
Multiple reviewers (TL8Y, VCmS) note that the long-context evaluation relies mainly on perplexity on PG-19 and does not test functional capabilities such as retrieval or question answering. The rebuttal does not substantially expand the evaluation beyond perplexity, leaving this concern outstanding.

**Reviewer Scores:**

- **Reviewer TL8Y (original score: 4, marginally below acceptance)**.
Given that the rebuttal addressed several technical questions (θ sensitivity, YaRN comparison, δc visualization) but did not resolve the core efficiency concern (“serious practical barrier” due to K/V cache overhead), I expect this reviewer would likely maintain a score around 4, possibly remaining marginally below the acceptance threshold.

- **Reviewer Y9wq (original score: 2, reject; confidence 5)**.
The rebuttal strongly disputes the novelty critique but does not clearly incorporate or reconcile the cited prior work, nor does it add efficiency evidence. Given the reviewer’s high confidence and central focus on novelty and citation issues, I believe this reviewer would not materially change their score and would remain at 2 (reject).

- **Reviewer LYyY (original score: 4, marginally below acceptance)**.
The rebuttal provides clearer theoretical positioning (PoPE as a more constrained inductive bias), adds YaRN comparisons, and responds to δc-related questions. These responses may increase the reviewer’s confidence slightly; however, the lack of formal analysis and broader benchmarks likely remains. I estimate this reviewer might move to a weak 4 or borderline 6, but still with reservations.

- **Reviewer VCmS (original score: 6, marginally above acceptance)**.
This reviewer already viewed the work positively but expressed interest in broader evaluations and efficiency considerations. Since the rebuttal does not fully address these points with new evidence, I expect this reviewer would likely remain at 6, maintaining a mildly positive stance.

---

### Decision · Program_Chairs · 2026-01-26

Reject